# Conventional Event Tree Analysis on Emergency Release of Liquefied Natural Gas

**DOI:** 10.3390/ijerph19052961

**Published:** 2022-03-03

**Authors:** Tomasz Zwęgliński

**Affiliations:** Internal Security Institute, SGSP (The Main School of Fire Service), 01-629 Warsaw, Poland; tzweglinski@sgsp.edu.pl

**Keywords:** liquefied natural gas, training, exercises, emergency response, event tree analysis

## Abstract

Liquefied natural gas (LNG) is stored in facilities located in urban areas and transported over public roads. A shift towards the broader use of LNG is economically and environmentally justified. However, it triggers an obvious need for an investigation of LNG risk through elaboration and validation of potential scenarios that the hazard might generate if it materialises. This background knowledge and past experiences were elaborated in the course of a literature review and field experiments towards designing the conventional event trees on LNG emergency release for three different units, such as a storage tank, a pipeline and in road or railway transportation. The research allowed us to answer the following question: what are the key scenario lines LNG incidents might follow? Thus, it constitutes a valuable tool for designing, planning, organising, executing and evaluating trainings and exercises on LNG emergencies.

## 1. Introduction

Liquefied natural gas (LNG) is becoming a significant source of energy. This tendency is broadly recognised in Europe and widely around the world. One of the highly visible signs for the expanding role of LNG in the energy market, especially in road transport [1], is the increasing volume of imported LNG into European Union states with sea carriers. Between 2015 and 2019, global LNG trade expanded by 45%, posting record growth in both 2018 and 2019 [2]. Hence, seaports have recently been intensively developing their capabilities in order to accomplish the growing needs and expectations of the European market in this respect. There is an increasing number of new LNG sea terminals, which, in many cases, constitute critical infrastructure facilities. The proximity to the market and consumer plays an important role due to the fact that LNG evaporates easily, and it is not economically justified to transport further than 300 km by land [3]. However, as LNG carriers reach the seaports, the cargo is offloaded to LNG storage tanks or degasified to be transported by a pipeline system. From these “entrance gates”, LNG is distributed all over Europe by pipeline as well as by railway and road transport, which are often qualified as critical infrastructures [4].

Basically, LNG has quite a lot of advantages, which determines its increasing usage:It is a cheaper source of energy compared to “black products” (petrol and diesel) and liquefied petroleum gas (LPG) [5];It is suitable for consumer purposes, e.g., in such fields as fuel to drive buses, trucks and cars, heating and cooking in households, in industrial processes like steel, paper and ceramic production, and also other sectors like tourism [6,7];It possesses a high energy content; therefore, the total energy needed is reduced [8];It is a “clean fuel” contributing to improved product quality, reducing maintenance costs whilst being more environmentally friendly compared to coal, diesel, petrol or LPG [9];Large amount of natural gas can be stored and transported at low pressures [10].

Besides LNG advantages, it also has some shortcomings that might potentially lead to the development of hazardous scenarios with variable impacts in character and range [11,12,13]. Such scenarios will naturally be determined by the physicochemical properties of LNG. Hansson [14], Er [15], and many others define LNG as a natural gas liquefied under a low temperature mixture (cooled to 162 °C). Following Ducommun [16], it is mainly composed of methane (from 85% to 99%) but also ethane, propane, butane and pentane, as well as carbon dioxide, helium, nitrogen and hydrogen sulphide. Zwęgliński [8] says that the boiling temperature is generally from −162.2 °C to −161.5 °C. LNG is odourless, colourless, non-toxic, non-corrosive and lighter than water. At high concentrations, the oxygen content in the air is reduced below the level necessary for life (below 12.5% of oxygen in the air). LNG is a cryogenic and flammable liquid. Therefore, the characteristics of LNG surely define the key hazards related to potential incidents with its presence.

Another aspect to be taken into consideration while designing LNG-related scenarios is the context of an incident. Such incidents might be elaborated for LNG storage tanks [17], pipelines, railway or road transportation [18]. As already argued, there are increasing quantities of LNG in use all over Europe, including LNG transportation. LNG is also more often used as fuel, mainly for trucks and city buses—this tendency will increase. Planas-Cuchi et al. [19], Vollmacher and van Esbroeck [13] as well as Klaos and Kriisa [3] proved that road incidents of trucks carrying LNG have already taken place, generating different types of impacts. For example, Alderman [10] states that an explosion scenario for LNG is possible only in confined spaces like transportation cisterns, especially in rollover conditions—however, it is impossible in open spaces. There are other objects like pipelines or storage tanks [20], in which various options of risk development have to be taken into consideration, including pool releases or different types of fires. Hence, broader research on potential LNG incident scenarios is postulated by many researchers and practitioners, such as Wellman et al. [21], Bralewski and Wolanin [22], Gyenes et al. [23] and others.

### Objectives

All in all, economically and environmentally justified shift towards broader use of LNG triggers an obvious need for investigating LNG risk through elaboration and validation of potential scenarios that the hazard might generate if materialised [24,25]. Thus, the objective of the research is to demonstrate potential adaptation of conventional event trees as a useful and feasible tool for disaster managers and first responders learning and training purposes in haz-mat incidents involving LNG [26,27,28,29,30]. Background knowledge and past experiences were elaborated in order to fulfil the paper’s aim, which is designing conventional event trees on LNG emergency release for such cases as storage in a tank or road and pipeline transport. The designed scenarios are, to some extent, conditional since the concept includes emergency response measures that might be undertaken during different phases of scenario development [13,16,31]. The efficiency of these measures, being implemented by first responders, determines the path of a scenario development at the end. Thus, the question answered with this paper is as follow: what are the key scenario lines the first responders have to acknowledge while being trained or responding to LNG incidents?

The novelty of the research is in proposing a concept for first responders and disaster managers’ competences development through adaptation and utilisation of conventional event tree methods tailored for training purposes, e.g., by using the “decision points” dilemma.

The results presented in the paper are to be taken up and implemented by first responders and disaster managers. Furthermore, it may be utilised as a supporting scheme for pre- and post-incident analyses.

## 2. Materials and Methods

A review of published research was used to identify, analyse and synthesise the knowledge of LNG properties. For the review purpose, the following keywords were used: “LNG”, “liquefied natural gas”, “LNG safety” and “LNG fire”, alone or in combination. The acquired data from Academia.edu and ResearchGate were organised, categorised and mapped. Next, abstracts of the revealed publications were reviewed to select those which dealt with LNG road, railway and pipeline transport, as well as storage. The selected pool of publications was furtherly limited by a full-text review for those which only concern the research devoted to real case incidents, analyses, exercises and simulations of LNG risk scenarios [32]. In addition, existing standard operating procedures, emergency response protocols and reports from LNG incidents, including trucks accidents, were analysed.

The results of the literature review enabled us to describe, in a synthesised manner, the key properties and characteristics of different potential scenario lines, including no fire, fire and explosion. Analysing these properties and characteristics, the event trees were designed. There was also a series of field tests conducted to enable observation of LNG behaviour in case of no fire and fire scenarios. These observations also provided information to the process. The field tests were carried out at the Field Training and Rescue Innovation Centre of the Main School of Fire Service (SGSP) in Warsaw (Poland). The experiments included an actual release of LNG from a truck tanker in order to observe how LNG disperses in changeable atmospheric conditions, how it reacts while ignited in different moments of the cloud and pool formation generating different types of fires.

The knowledge from the literature and protocol review was collected, analysed, synthesised and finally compared with the observations captured in the field experiments. The results were used to design conceptual trees of LNG scenarios. Furthermore, the concept was reviewed and discussed during expert workshops organised in frames of the scientific project the research is done for. The analytical part was followed by a synthesis. At this stage, on the basis of the collected data during the review process, the concept was verified and adjusted to the final form, including key scenario lines for three different LNG objects: cistern, pipeline and storage tank. The conditional character of the concept was determined by the fact that the scenario line should be taken into account while analysing the conventional event trees, and highly depends on the effectiveness of decisions and broader emergency response measures undertaken during a full-scale exercise or drill (real first responders’ operation during an exercise), table-top exercise (decisions taken on a certain phase of scenario development, e.g., for education, training, pre-incident analyses) or potentially real LNG incidents (e.g., for post-incident analyses).

The concept is mainly aimed for utilisation by incident commanders and first responders during training or exercises. However, to make it work, the collected data were synthesised to a consistent and coherent diagram. Thus, such a synthetic concept has been produced to ensure an easy uptake, digestion and adaptation of the research results by first responders for their daily work, including training, exercises and potentially pre- or post-incident analyses.

## 3. Background Information

Utilisation of conventional events trees for risk assessment purposes is a common and well-known method. It is broadly used in technical sciences; however, the literature review results show that it is rarely utilised for first responder and disaster manager training purposes, specifically in the context of LNG incidents. Some work in this respect has been presented by the Joint Research Centre [23] that has been a trigger point for starting the research presented in this article. Therefore, it is the extension and adaptation of the background knowledge present in the literature for first responders and disaster manager training purposes.

By 2004, only nine accidents involving liquefied gas vehicles in road transport were recorded in the Major Hazard Incident Data Service (MHIDAS) [33]. In four of these cases, there was no leakage or fire; in three cases, there was a gas leak where ignition was prevented. In one case, car tyres and fuel ignited, but the contents of the tank remained untouched by the fire. Only in one case (Spain 2002), a cargo of LNG caught fire, killing one and injuring two people. Therefore, the elaborated concept focuses on the two main alternatives of LNG release, which are dispersion without or with ignition source presence. The third option considered is a sub scenario for dispersion with ignition, which can lead to an explosion.

### 3.1. Properties

LNG is a mixture of various gases, mostly comprising methane (usually 95–97%) and small amounts of ethane, propane and other heavier compounds. LNG is not a standard mixture, so the composition of each specific natural gas mixture depends on the manufacturer. The density and boiling point of LNG also varies slightly from one mixture to another. The boiling temperature is generally from −162.2 °C to −161.5 °C. Specific hazards associated with LNG are related to its physical and chemical properties [19]. As a result of the reduced oxygen content, dyspnoea occurs, which starts with symptoms of drowsiness, fatigue and loss of coordination and can lead to a loss of consciousness. Under some conditions, when gaseous, it can be invisible to humans, which also makes it dangerous. This depends on the air humidity level. Invisible gases can cover surroundings beyond the visible cloud of LNG. The natural gas reaching the end-user is vaporised, but only before it is distributed to the consumer. Transported methane is odourless, so special gas detectors should be used during transhipment and rescue operations [34,35]. The liquefied gas cannot be rehydrated because the liquefied gas must be free of pollution. Extremely high and non-flammable gases (not ignited) are the main hazards of the extremely low temperature of the gases, which is harmful in contact with both living tissue and equipment.

### 3.2. Dispersion

When released on a large scale, LNG is, for some time, a mixture of air and gas, which is denser than the warm ambient air. The area covered by a cloud in the case of a weak wind or temperature inversion can be quite large and some distance away from the leakage site if the concentration of the LNG in the mixture remains below the lower explosive limit. As the gas continues to heat and the temperature rises above −110 °C, it becomes lighter than air and begins to disperse in the atmosphere.

The boiling point of LNG compared to the ambient temperature is very low (−162 °C), which causes very fast evaporation after a leak. There are test results showing that if 10,000 tonnes of LNG are released into the aquatic environment, the total evaporation time will be about 5 min [36].

There are several factors that influence how the gas disperses in the atmosphere. This is influenced by the weather conditions (wind, temperature and humidity). Differences in these parameters change horizontally and vertically over time. In addition, the ground roughness coefficient, the composition of the gas cloud and a number of other small factors influence the process as well. LNG vapour, when mixed with air, heats up simultaneously. The movement of the gas mixture according to the wind direction is described as long, low and thin clouds shaped like cigars or feathers.

The most important hazards of LNG are frost damage caused by cryogenic features (freezing of skin, equipment, etc.), ignition and explosion. When the liquefied gas leaves the tank, it immediately starts to heat up and becomes gaseous. First, the gas produced is heavier than the surrounding air, creating a cloud of steam over the flowing liquid [37].

One of the most unique properties of liquefied gas is its extremely low temperature. During evaporation, dynamic cooling of the environment occurs, which is a threat to living matter but also to most metals which, when frozen, become brittle and more easily damaged (reduced impact resistance). Rubber and many metals (e.g., steel) tend to lose their properties at such low temperatures (they become brittle). Various devices (pressure hoses and pipes), steel structures and welds (valves, fittings, etc.) are the weakest elements, especially when subjected to force in such conditions.

LNG, being a cryogenic substance, is dangerous for people due to the extremely low temperatures in which it is transported. If a person is exposed to low-temperature gas in the event of a leak, serious frostbite or freezing may occur. Skin and lungs are exposed by inhaling cold fumes. Personnel entering the danger zone must wear appropriate protective equipment, which should protect the body and respiratory tract from extremely low temperatures and limited oxygen concentrations in the LNG cloud. In contact with liquefied gas or any metal cooled by the LNG, damage to skin tissue occurs more quickly than if it were in contact with the gas. In case of actual exposure, it should also be remembered that cryogenic fluids are substances of very low viscosity, which means that they easily penetrate the pores of the skin tissue as well as through clothing.

### 3.3. Fire

Fire was identified as another risk connected to the usage of LNG. Besides causing freezing and suffocation, a LNG cloud or pool can ignite and result in associated fire. Compared to other flammable liquids, LNG vapour has a relatively narrow explosive range. The lower and upper explosion limit at 25 °C is 5–15%, respectively. If the gas concentration is less than 5%, it cannot ignite due to the lack of sufficient flammable material. If the gas concentration is higher than 15%, it cannot ignite due to insufficient oxygen. To ignite LNG, it must be released from a tank, evaporated, mixed with air to form flammable concentrations and then brought into contact with an ignition source [3].

Many researchers have speculated that a gas cloud could ignite with every large spill, but this is unlikely to happen [38]. On the one hand, this is justified by the rather narrow explosive limits, and on the other hand, the heated gas mixture is much lighter than air and disperses in the atmosphere when the temperature rises rapidly. However, the flammable product may remain at a low temperature, in which case, depending on the weather conditions, the gas mixture will move quite low above ground for a long time. Due to the low ambient temperature, the vapour behaves like a cloud of gas heavier than air, forming water vapour in the air through condensation visible as a white mist.

During a rescue operation, the risk assessment must take into account that at humidity levels above 55%, a gas cloud that may be within its explosive limits is visible. At humidities lower than 55%, a gas cloud with a concentration within its explosive limits may not be visible. One of the most important indicators in the case of a flammable gas leak is the spread of the cloud as a function of distance. With distance, the probability increases that the concentration will fall below its lower explosive limit. The further away the gas cloud is able to spread (within the flammable limits), the greater the chance that it will ignite. In a generalised form, the flammability can be assessed by following practical rules, which are used as a basis for risk analysis. If the lower flammable limit is less than 100 m from the leakage point, the probability of ignition is 5%. At an altitude of 101–300 m, it is already 30% there and up to 95% at over 300 m. When a gas cloud passes through the wind in an industrial or residential area, you can be sure that it will come into contact with an ignition source, causing a fire in the gas cloud and a fire spreading back towards the leakage point. In the case of large clouds of natural gas, a gas cloud may pose an ignition risk with a maximum range of 1600 m. In exceptional cases, e.g., in the case of a very large leak from a tank at an offshore terminal, this size should be extended 2–3 times [3]. When a large and unrestricted puddle of liquefied gas at the leak site is ignited, it burns until it burns out completely. In this case, it is often not possible to extinguish a large spill unless the flow of combustible material can be stopped. If a gas cloud is ignited at a certain distance from the leak, combustion will take place in the direction of the leak. The propagation rate of combustion is not high and is comparable to walking speed (about 2 m/s). The combustion of LNG usually takes place very quickly, and the heat generated in this process is very intense. Injuries resulting from the combustion of LNG can be as follows:By heat radiation;By direct contact with the flame;By inhalation of hot combustion products.

In a situation where a non-flammable gas cloud has started to spread to a populated area, an informed decision whether or not to ignite must be taken to ensure greater safety [13,34,38,39]. If the inhabited area is so far away that there is a gas cloud if it reaches the ignition limits, there is a very high risk of unintentional ignition. In such a case, the gas cloud should be ignited and sacrificed proactively along with the leaking plant. The main weighting point is the possible consequences for different scenarios—the past additional risks caused at the ignition site or ignition in a populated area [40]. Unfortunately, there are no specific studies on the liquefaction dispersion of the gas cloud and its subsequent ignition.

### 3.4. Explosion

An explosion is another risk worthy of elaboration. It may occur if the substance is released uncontrolled and ignites. Such a release may occur as a result of a rupture or damage of the pressurised container. Taking into account that LNG is stored in tanks under atmospheric pressure, damage to the container structure should not cause an immediate explosion. Natural gas is reactive, and therefore, explosions of mixtures with hydrogen, propane or ethylene are possible. The risk of explosion is greater when the gas cloud is in a limited or enclosed space. Heating of containers is associated with a high risk of explosion. In this situation, rapid evaporation will occur as the temperature in the container rises, and an increase in pressure leads to the container breaking up. As a result, the entire LNG content is released, creating a large cloud of flammable gas.

In general, different authors exclude the immediate failure of a container due to accident or explosion (BLEVE—boiling liquid expanding vapour explosion) in the transport of LNG because the gas is not transported under pressure. Risk of explosion is based on the assumption that the gas is transported under atmospheric pressure, which is not the case with LNG. However, LNG transported by road in Estonia, for example, is under pressure, and the gas stored in tanks, therefore, remains liquefied above its normal boiling point [3]. Pressurised containers create greater risks than usual, including the risk of explosion. Similar conclusions were drawn in 2002 after the incident that occurred in Spain. There was an explosion with signs of BLEVE. As a result, it was concluded that LNG is not so different from other flammable liquids because of its specific properties, so you can not completely rule out BLEVE, especially if the truck cistern rolled over during the accident. Although the accident in Spain was rather exceptional, it changed the way of thinking about how to deal with the risks associated with LNG.

## 4. Results and Discussion

Figure 1, Figure 2 and Figure 3 present the conventional event trees in three different versions reflecting potential LNG incidents in critical infrastructure objects playing a key role in the gas distribution. There are three cases elaborated: road and railway transport, a storage tank and a pipeline. These cases are elaborated towards three crucial scenario line developments, which are dispersion without fire, dispersion with ignition and explosion. Each of the trees reflects these three potential scenario lines, determined from the literature review and field tests. Real cases of such incidents might be found in the referenced sources to the article, e.g., [3,8].

Moreover, in these trees, a particular scenario line flow is conditioned by incident commander decisions and the effectiveness of emergency mitigation measures introduced on certain phases of the overall scenario development. In this respect, the orange boxes reflecting “decision points” and the yellow boxes reflecting effective emergency response measures (“learning curves points”) are essential. These elements of the trees are conditional and depend on the decisions and performances of trainees during the training/exercise being evaluated by trainers. Solid lines depict more likely scenario flow, while the dotted ones suggest less probable development (red dotted line reflects low probability high impact sub scenario).

Each combination of the scenario flow is ended with a generalised assessment of potential impact generated by the analysed incident. The categorisation of the impact is formulated on the basis of the literature review presented in the Background information section; however, it is definitely to be underlined that the impact strongly depends on the hazard (e.g., volume of LNG), exposure (the character of the vicinity of the incident, e.g., industrial, urban, etc. as well as weather conditions) and vulnerability of the exposed people, infrastructure and natural environment. As such, the suggested impact is, to a certain extent, generalised.

LNG conventional event trees are designed and shaped in order to be used for training and exercising purposes. Primarily, they can facilitate designing and creating scenarios for such didactic events. However, the key value of the concept is the ability to use ‘decision points’ (labelled orange) and ‘learning curve points’ (labelled yellow) for active learning and training purposes. The overall idea is that the abovementioned moments of a training or exercise are key moments when trainees get to actively interact with their trainers in a table-top event. This interaction is devoted to recognising the trainee’s knowledge and skills on a specific topic related to the particular moment of the scenario flow. However, if the didactic event is a full-scale exercise/training or drill, by analogy to the table-top, the trainers are obliged to closely observe the performance of the trainees in the given points of the scenario flow. This needs to be done in order to assess the trainees’ competencies which, in the case of the table-top, is done in the form of a discussion. On the basis of the acquired data by discussion and/or observation, the trainers decide on the optimal path of the further development of the scenario. The trainers’ decision is obviously related to the aims of the training/exercise but also on some other aspects like the level of performance of the trainees in a ‘decision point’ or in a ‘learning curve point’, the level of difficulty the training/exercise is expected to force on the trainees, the optimal learning path for a given trainee in the given context of his/her mental and physical disposition on that particular day. Such an approach guarantees a certain level of flexibility stimulated by the trainers, which positively influence achieving optimal learning outcomes for the given trainee on a given day. 

Methodologically, training is conducted by specialised trainers on the basis of the abovementioned ‘decision points’ and ‘learning curve points’. The methodical assumption allows them to verify existing, and to form new, qualifications of the trainees in terms of their knowledge, skills and social competencies (e.g., by discussion during table-top). The latter are mainly due to the possibility of conducting the training in a team setting, reflecting different roles of the trainees, e.g., incident commander, chemical rescue team leader, rescuer, etc. The ‘decision points’ and ‘learning curve points’ are didactic pauses during training to allow contact with trainers for problem-solving discussions at key moments in the scenario development. Hence, the trainer may discuss corrective actions with the trainees when they realise the trainees acted inadequately regarding the situation and the standard operating procedures. The same relates to a situation while the trainees decide on a response measure, which is not acceptable at the given moment of the scenario and its context, from a risk assessment perspective. All such learning curves are recommended for in-depth elaboration between the trainers and trainees. Regardless of the form of training, as well as the setting of the simulation and gameplay (in stationary facilities, e.g., simulation with film or in field conditions with gas or LNG simulators), the ‘points’ allow decisions and planned actions in the face of the situation arising from the scenario at a given moment of its development to be determined with the trainees. On the basis of these decisions, made depending on the sophistication of the trainees with more, less or no support from the trainers, the trainers decide on the further path for the training to take according to the scenario schemes developed. Thus, the training is characterised by open scenarios. The final variant of the applied scenario path during a given training/exercise depends on the trainers and their evaluation of a number of decisions made by the trainees at each ‘point’. 

The concept has the didactic value of interaction with the trainers at different stages of the training/exercise. In addition, the pace of training implementation is dictated by the trainers based on ongoing monitoring and evaluation of participants’ decisions, actions and behaviours. It also allows trainers to adjust the difficulty level of the scenario to the progress of the participants during a given training flexibly and on an ongoing basis, for example, by increasing or decreasing the didactic pressure on them (limiting the time for decisions, introducing additional variables to the problem situation, e.g., more victims, spills dynamic and structure, limiting the availability of resources at the scene, e.g., due to delayed arrival of a specialised chemical rescue team, etc.). Despite the freedom of trainers in the variation of the scenario, allowing for their adaptation to the training/exercise purposes and aims, and the participants’ condition on the day, the main scenario paths remain constant. This is a prerequisite, as previous research has shown that there are several main scenarios for the development of LNG accidents, which are foreseen in the developed schemes presented in this paper. 

The proposed didactic form activates the trainees at each stage of the scenario, which positively influences their achievement of the desired learning outcomes. In addition, it enables the active formation of new knowledge, skills and social qualifications and allows the trainee to review and revise existing competencies. This is made possible by a reflective self-assessment by each participant at scenario ‘points’ with the mentoring support of the trainers. The level of support offered by the trainers at the ‘points’ should correspond to the learning goals and objectives of the training/exercise as well as to the pre-reception level of the trainees. 

It is also worth emphasising that ‘points’ allow trainers to collect the necessary data to evaluate the didactic progress of the participants. Depending on the training objectives and the profile of the trainees, the trainers may place more or less emphasis on a certain range of rescue activities during a specific training/exercise session. The purpose of the concept is to quantitatively verify whether the trainee has covered the optimal range of competencies assigned to him/her in the realisation of the objective function, which is to minimise the risk of catastrophic consequences of a LNG event (according to the scenario diagrams developed in this paper). Moreover, in qualitative terms, trainers assess whether and how a given participant, as well as the entire training team (e.g., tactical union in the form of a platoon, specialist rescue group or various other configurations), applied functioning procedures, rules, good practices, as well as what decisions were made at a given ‘point’ by persons in charge of the rescue operation (e.g., incident commander, chemical rescue team leader). To this end, the trainers will, depending on the objectives of the training/exercise, formulate their expectations for decisions, rescue actions and measures to be taken by the trainees at the ‘point’ during the development of the evaluation method. The reference material for the trainers to develop the abovementioned requirements should be constituted by legal regulations, internal regulations of the rescue system (e.g., guidelines), but also the so-called good practice developed through previous experience and knowledge of the trainers. It is proposed that the training/exercise introduction, as well as didactic materials, should be available for upcoming trainees before the training/exercise to enable them to prepare for it theoretically, e.g., in the form of self-study or e-learning [41].

Apart from having a well-founded professional knowledge and skills in this area to evaluate the content of the participants, trainers should be methodically prepared for the task of evaluating the participants. Besides solid competencies and experience in the field of chemical rescue, the trainers should acquire their methodical competence through training, workshops or other didactical forms on methods and techniques of evaluation of trainees. Such training should contain contents concerning the evaluation process as an object of cognition. Thus, it should include basic mechanisms and phenomena inherent in the theory of cognition, taking into account elements of psychology, including social psychology (e.g., “two systems of thinking”, “cognitive errors”, etc.), especially in the context of evaluating people and phenomena. Moreover, the training should lead to trainers’ acquisition of so-called soft competencies in the area of maintaining proper mentoring. Trainers should have an authority based on their competencies and attitudes towards trainees. Methodologically, the training should provide trainers with the knowledge and skills to use appropriate methods, techniques (e.g., OAJR—observe-analyse-judge-recommend) and evaluation tools to objectivise the evaluation process (e.g., using at least two trainers to evaluate the same aspects) [42]. In general, the evaluation should provide information in three areas:The didactic progress of the trainees;Strengths and weaknesses of LNG response, including suggestions for improvement [34,38];Opportunities to improve the training method, including the evaluation method, for further development and improvement of the training/exercising concept.

Thus, the primary task of the evaluation process is to verify the level of learning outcomes achieved by individual trainees. Moreover, the evaluation should provide the necessary feedback from the trainees on how to improve this form of training and how to evaluate the trainees. Such an approach, i.e., the use of methodically grounded stages of training organisation, starting with the definition of its objectives, then evaluation methods adjusted to these objectives, and further organisational steps, will provide a logical and organisational continuum, both with regard to one specific training, as well as the whole, self-improving system, of training and rescue [43].

It is worth noting that the concept allows for feedback from the participants on the level of preparation and execution of the training/exercise, which fits into the third proposed evaluation objective. This makes the concept open for continuous improvement of the training/exercise content and forms.

Another advantage is its versatility and, therefore, the possibility of implementation within practical staff and application exercises.

## 5. Conclusions

The proposed conventional event trees for LNG emergency release in three different units is a concept that is based on the physical and chemical properties of LNG. It has been designed with deep consideration and analysis of past experiences and real cases of LNG incidents. The idea of having training checkpoints, such as ‘decision points’ and ‘learning curve points’ in the continuum of a training or exercise enable the achievement of optimal learning objectives in a flexible way which take into account the trainee individual mental and physical disposition on a day, scalability of difficulty level as well as realistic scenario paths determined by the performance of the trainees facing concrete operational problems. As such, it might be used for different types of exercises starting from command post, table-top, up to full-scale field exercises with real use of LNG or its simulants.

The concept has some potential to be further researched and developed as well as to be used for other purposes than presented in the paper, e.g., for post-LNG incident analyses. In addition to that, based on the past incidents, current knowledge and exercise experiences in this field, the proposed approach might be further adapted and developed to more advanced traditional event tree analyses, including calculating the risk of individual sub scenarios. This would mature the concept and perhaps make it feasible to be implemented as a scenario engine in IT-based didactic tools, e.g., virtual simulations, gamification, etc.

## Figures and Tables

**Figure 1 ijerph-19-02961-f001:**
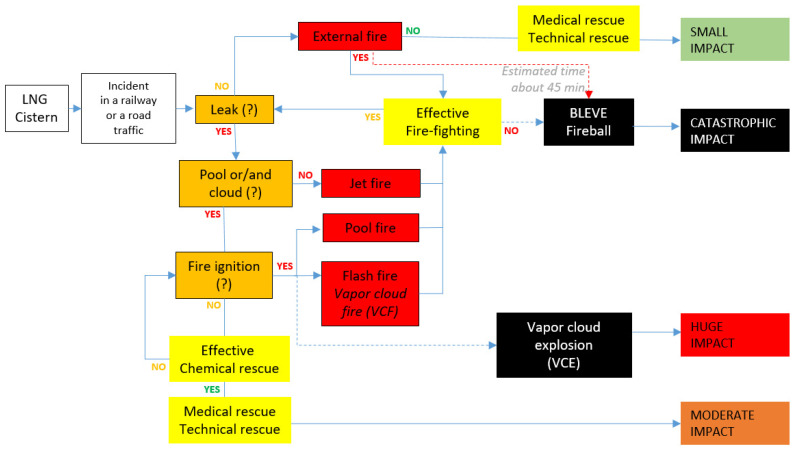
LNG conventional event tree for road and railway transport.

**Figure 2 ijerph-19-02961-f002:**
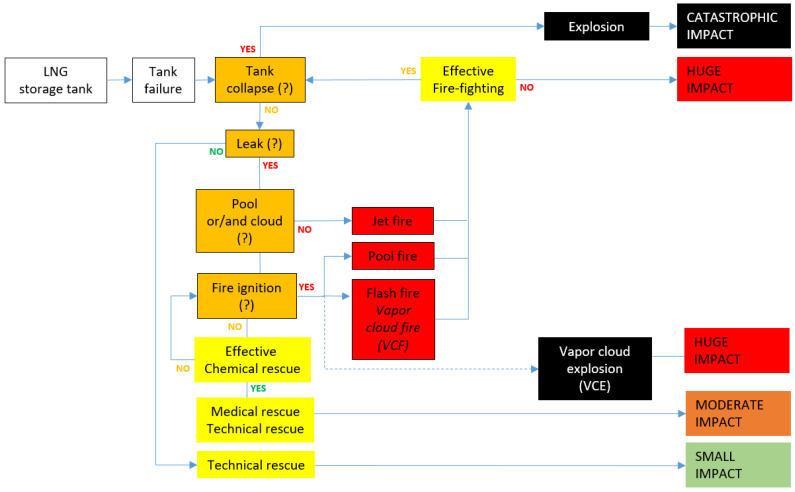
LNG conventional event tree for a storage tank.

**Figure 3 ijerph-19-02961-f003:**
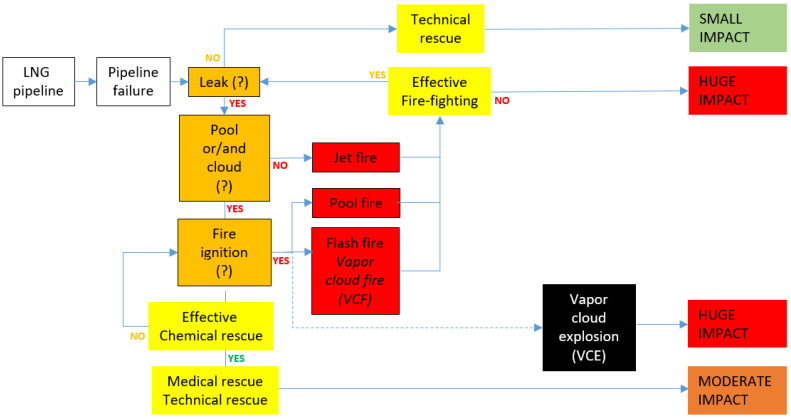
LNG conventional event tree for a pipeline.

## Data Availability

Data sharing not applicable. No new data were created or analyzed in this study. Data sharing is not applicable to this article.

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
