# Peer review of "Conventional Event Tree Analysis on Emergency Release of Liquefied Natural Gas"

_ijerph, 2022, doi:10.3390/ijerph19052961_

Round 1

Reviewer 1 Report

The topic chosen by the author is extremely important and topical in economic and environmental terms is closely related to domestic and international scientific discourses. The author presents a number of concepts and approaches that are well-known to those skilled in the art in both domestic and international scholarly discourses.

The abstract is sufficiently clear and adequately summarizes the content of the article. Keywords are correct. But it would be worthwhile to include event tree analysis as key words.

The areas covered by the author are closely related to the issues of the safe use of liquefied natural gas, and their elaboration is sufficiently thorough to the level of a scientific article. The processing of the literature on the topic is appropriate in terms of content and form. The article meets the requirements of the journal stylistically, professionally and formally. However, the study could be even better if the author elaborated on the method used in his study, the traditional event tree analysis. It would also be important to describe in detail the main steps in applying this method and the main advantages and limitations of the method.

The demarcation of the research question is clear. The author's problem statement and scientific contribution can be clearly delineated. The article provides a theoretical or practical contribution along a clear research to a deeper understanding of the topic introduced. The main virtue of the article is that the author apparently worked on the basis of a well-thought-out research concept, so it can be evaluated as a thorough scientific work.

The article analyzes the research questions and problems it raises in an adequate way, its guidance is clear and logical. The principles of scientific reasoning and proof are at work, but at the same time they are simplistic and understandable to the widest possible audience.

The article contains clearly articulated results, either for practice or for scientific theory. However, the delimitation of the results is not yet clear, it would be worthwhile to go into more detail than at present on the possible and proposed corrective actions to be taken if the outcome risk of a path is not acceptable. It would be useful to explain the directions of further investigations in more detail than at present.

Author Response

Dear Reviewer,

First of all thank you very much for your positive feedback as well as suggested corrective measures. These hints are highly appreciated and indeed draw my attention to the points I have not realized before. Therefore, I addressed Your points as follow:

“…it would be worthwhile to include event tree analysis as key words.”

The key word has been added. Thank you.

“…the study could be even better if the author elaborated on the method used in his study, the traditional event tree analysis.” + “..It would be useful to explain the directions of further investigations in more detail than at present.”

I addressed these two points in a following way. In Conclusions I have proposed that the concept may be furtherly developed, including calculating risks of individual sub scenarios in the traditional event tree analysis. I have also suggested that having it researched might provide essential data for implementation of the concept to IT driven didactic tools.  

“The article contains clearly articulated results, either for practice or for scientific theory. However, the delimitation of the results is not yet clear, it would be worthwhile to go into more detail than at present on the possible and proposed corrective actions to be taken if the outcome risk of a path is not acceptable.”

This aspect I addressed in more than one place of the article. In Results section I tried to cover it by adding “In this respect essential are the orange boxes reflecting ‘decision points” and the yellow boxes reflecting effective emergency response measures. These elements of the trees are conditional and depends on the decisions and performance of trainees during the training/exercise being evaluated by trainers”. Furthermore in Discussion I have added “Hence the trainer may easily discuss with the trainees corrective actions to be taken while he/she realizes the trainees decide inadequately to the situation and standard operating procedures. The same relates a situation while the trainees decide on a response measure which is not acceptable at the given moment of the scenario and its context from risk assessment perspective. All such learning curves are recommended for in-depth elaboration between the trainers and trainees.”

Once more thank you very much for highly valuable contribution and critical viewpoint. I do hope I covered all the suggestions in an acceptable way from the review standpoint.

Regards

Tomasz

Reviewer 2 Report

The author's research on liquefied natural gas is very important in urban life. This helps to improve the safety management of the city. In this study, three conventional event tree deduction processes are proposed and introduced. It is suggested that the author can improve this paper by combining some real LNG treatment events. More specific deduction will help to achieve better training results.

Author Response

Dear Reviewer,

First of all thank you very much for your positive feedback as well as suggested corrective measures. These hints are highly appreciated and indeed draw my attention to the points I have not realized before. Therefore, I addressed Your point as follow:

“It is suggested that the author can improve this paper by combining some real LNG treatment events. More specific deduction will help to achieve better training results.”
Agree. It would be wonderful, however there is quite limited number of reports on real cases available. To some of them I have already referred in the article. As shown in the paper there are not many LNG incidents in general, and even less is somehow elaborated in literature. Therefore, at least I tried to address them in the text (providing references to source materials) in the current “Background information” section. I also address this point I the text in the current Results and discussion section.

Once more thank you very much for highly valuable contribution and critical viewpoint. I do hope I covered all the suggestions in an acceptable way from the review standpoint.

Regards

Tomasz

Reviewer 3 Report

Comments:

Overall, the manuscript needs major review and restructuration. Please try to improve the overall document and its content.

Abstract:

  • Briefly include the main findings of the research.

Keywords:

  • Use “Liquefied natural gas”, instead of LNG.

Introduction:

  • Avoid terms as “more and more” (line 22)
  • Line 25 “is the increasing volume of imported LNG into European Union states 25 with sea carriers. “ - reference? How much did it increase and in how many time?
  • Lines 36 to 49 – references for each point?
  • Clearly state in the introduction (as using section 1.1. for example) the objectives, novelty and contributions of the work, that are not clear in the introduction.

Material and methods:

  • I believe Web of Science or Scopus would be more suited as a data source, instead of academia and research gate, but it is too late now (please consider it for your next manuscript).

Theory:

  • This section should be placed after introduction and materials and methods, or materials and methods should be a subsection from the introduction, and then “Theory” should be called “Literature review” or “background information”.
  • Inclusion of a brief state of the art instead of only literature review.

Results:

  • Figure 1, 2 and 3 are in an unacceptable format concerning their quality.
  • Result section only has 7 lines?

Results/Discussion

  • As the results section is not presentable in its current form, please compile the previous with the discussion section “Results and discussion”.

Conclusions:

-The conclusions must be improved, it is very weak in its current form.

Author Response

Dear Reviewer,

First of all thank you very much for your positive feedback as well as suggested corrective measures. These hints are highly appreciated and indeed draw my attention to the points I have not realized before. Therefore, I addressed Your points as follow:

“…Overall, the manuscript needs major review and restructuration.”

The review and reconstruction, as suggested in one of the below points, has been done. The sections are replaced accordingly. All review points are covered as described below.

“Use “Liquefied natural gas”, instead of LNG.”

Changed accordingly.

“Avoid terms as “more and more” (line 22)”

Changed.

“Line 25 “is the increasing volume of imported LNG into European Union states 25 with sea carriers. “ - reference? How much did it increase and in how many time?”

Filled with “Between 2015 and 2019, global liquefied natural gas trade expanded by 45%, posting record growth in both 2018 and 2019 [11].” Reference GIIGNL - International Group of Liquefied Natural Gas Importers.  The LNG Industry GIIGNL Annual Report 2019. Neuilly-sur-Seine 2019.

“Lines 36 to 49 – references for each point?”

As suggested each point referenced to individual literature source.

Clearly state in the introduction (as using section 1.1. for example) the objectives, novelty and contributions of the work, that are not clear in the introduction.

Additional subsection included in the Introduction. Objectives and other suggested aspects are more precisely specified e.g. “…the objective of the research is to demonstrate conventional event tree as an useful and feasible tool for disaster mangers and first responders learning and training purposes as it comes to haz-mat incidents involving liquefied natural gas.”

I believe Web of Science or Scopus would be more suited as a data source, instead of academia and research gate, but it is too late now (please consider it for your next manuscript).

Indeed.
The suggestion well noted and to be used for next articles of this type.

This section should be placed after introduction and materials and methods, or materials and methods should be a subsection from the introduction, and then “Theory” should be called “Literature review” or “background information”. Inclusion of a brief state of the art instead of only literature review.

Changed as suggested. The new title of the section is “Background information” which appears after Introduction and Materials and Methods. A paragraph on state of the art has been added.

Figure 1, 2 and 3 are in an unacceptable format concerning their quality.

Agree. If it was possible, I would suggest to leave the decision on re-formatting the figures to the editor, please. Counting on your understanding.  

Result section only has 7 lines?

As the results section is not presentable in its current form, please compile the previous with the discussion section “Results and discussion”.

Thank you for this suggestion. Fully respect and follow combining these sections.

The conclusions must be improved, it is very weak in its current form.
The Conclusions are extended with the focus on the objective of the article. Future research directions appointed.

Once more thank you very much for highly valuable contribution and critical viewpoint. I do hope I covered all the suggestions in an acceptable way from the review standpoint.

Regards

Tomasz

Reviewer 4 Report

The paper proposes the conventional event trees for emergency release of Liquified Natural Gas in three different units. The accidents involving LNG might have catastrophic consequences; thus, event trees might facilitate possible investigation and enable the implementation of corrective measures that might help mitigate the effects of such accidents, for example. I have a few comments and suggestions given below.

  1. Line 93 - It is stated that Academia.edu and ResearchGate databases were used for searching existing research papers including LNG properties. Why not use some other databases?
  2. Line 128 - it is stated: "By 2004, only 9 accidents involving liquefied gas vehicles in road transport were recorded in the MHIDAS database". Are there any newer data available in the database (18 years gap)? When first time mentioning some acronym (MHIDAS), please write it in full.
  3. In the Results section, three Figures are presented. Please elaborate on how were they developed (literature review and field experiments?).
  4. Figures 1, 2 and 3 include small impact, moderate impact, huge impact and catastrophic impact, but it is not elaborated on each of these. Please explain what constitutes each of these. How was it determined what, for example, means small impact?
  5. Are there any limitations of the research?

I hope that my comments and suggestion will help to improve your paper.

Author Response

Dear Reviewer,

First of all thank you very much for your positive feedback as well as suggested corrective measures. These hints are highly appreciated and indeed draw my attention to the points I have not realized before. Therefore, I addressed Your points as follow:

Line 93 - It is stated that Academia.edu and ResearchGate databases were used for searching existing research papers including LNG properties. Why not use some other databases?

Indeed, it would have been worthy to explore also other research platforms, however, not done unfortunately. The point was also noted by another reviewer. However as the reviewer rightly noticed it is not possible on this stage of the manuscript. The note is thoroughly taken and will serve me for next articles. Thank you.

Line 128 - it is stated: "By 2004, only 9 accidents involving liquefied gas vehicles in road transport were recorded in the MHIDAS database". Are there any newer data available in the database (18 years gap)?

To my best knowledge there are, however not publicly available and difficult to get access to them. Moreover, since I based on literature review (with the above mentioned limitations e.g. to the two research platforms), the data presented are the ones which I explored in this process. Nevertheless I agree that it would be lovely to get more updated information, however the statistics of LNG incidents are not changing dynamically and there are relatively small number of incidents like this.

When first time mentioning some acronym (MHIDAS), please write it in full.

Corrected.

In the Results section, three Figures are presented. Please elaborate on how were they developed (literature review and field experiments?).

Mainly it was a theoretical analyses on the base of the literature review, as stated in the Materials and Methods section. The results of the review are presented in the current “Background information” section where LNG properties are presented and the reasoning of the three types of releases (no fire, fire, explosion). Analyzing these properties and characteristics the event trees were designed. Moreover, I added in the text some elaboration on the starting point to the analysis which was the Joint Research Centre publication referenced in the article. On top of it there was a series of field tests conducted (also explained in the Materials and Methods) which enabled observation of LNG behavior in case of no fire and fire scenarios. These observations also provided information to the process.
Relevant additions were made in the section Materials and Methods.
Hopefully, this explanation makes it more clear now.

Figures 1, 2 and 3 include small impact, moderate impact, huge impact and catastrophic impact, but it is not elaborated on each of these. Please explain what constitutes each of these. How was it determined what, for example, means small impact?

Fully agree. I added some explanation on it in the current Result and discussion section. It is hardly to calculate it since it depends on many aspects, however on the base of the literature review and basic risk assessment (e.g. simulations) it is possible to generalize and at least initially assess/foresee an impact for each sub scenario.

Are there any limitations of the research?

Yes, I shortly referred to it above. The limitation is the scope of articles reviewed (e.g. in frames of two, not more, research platforms). These limitations I expressed in the Materials and Methods section.

I hope that my comments and suggestion will help to improve your paper.

Indeed. Thank you very much for your highly valuable contribution and critical viewpoint. I do hope I covered all the suggestions in an acceptable way from the review standpoint.

Regards

Tomasz

Round 2

Reviewer 4 Report

The paper is improved, and all my comments and suggestions are answered.